# ONC201/TIC10 enhances durability of mTOR inhibitor everolimus in metastatic ER+ breast cancer

Elena Farmaki, Aritro Nath, Rena Emond, Kimya L Karimi, Vince K Grolmusz, Patrick A Cosgrove, Andrea H Bild*

Department of Medical Oncology and Therapeutics Research, City of Hope National Medical Center, Duarte, United States

**Abstract** The mTOR inhibitor, everolimus, is an important clinical management component of metastatic ER+ breast cancer (BC). However, most patients develop resistance and progress on therapy, highlighting the need to discover strategies that increase mTOR inhibitor effectiveness. We developed ER+ BC cell lines, sensitive or resistant to everolimus, and discovered that combination treatment of ONC201/TIC10 with everolimus inhibited cell growth in 2D/3D in vitro studies. We confirmed increased therapeutic response in primary patient cells progressing on everolimus, supporting clinical relevance. We show that ONC201/TIC10 mechanism in metastatic ER+ BC cells involves oxidative phosphorylation inhibition and stress response activation. Transcriptomic analysis in everolimus resistant breast patient tumors and mitochondrial functional assays in resistant cell lines demonstrated increased mitochondrial respiration dependency, contributing to ONC201/TIC10 sensitivity. We propose that ONC201/TIC10 and modulation of mitochondrial function may provide an effective add-on therapy strategy for patients with metastatic ER+ BCs resistant to mTOR inhibitors.

*For correspondence: abild@coh.org

Competing interest: The authors declare that no competing interests exist.

## Editor's evaluation

The study by Bild et al. reported a valuable finding on the combination use of ONC201/TIC10 towards ER+ breast cancer. The data of the manuscript is rather solid. The paper may produce translational impact and scientists working in the field of breast cancer may benefit most from the work.

## Introduction

Breast cancer (BC) is a fatal disease with 287,850 new cases in 2022 in the United States and an estimated mortality rate of 43,250 (https://seer.cancer.gov/statfacts/html/breast.html). The most common BC subtype is positive for hormone receptor (HR+) and does not overexpress the human epidermal growth factor receptor 2 (HER2/neu), accounting for 68% of all BC cases (https://seer.cancer.gov/statfacts/html/breast-subtypes.html). The primary treatment for estrogen receptor positive (ER+), HER2− BC are therapies targeting estrogen signaling; however, resistance to these therapies in metastatic BC (mBC) and disease progression remains a significant challenge (*Osborne and Schiff, 2011*).

Aberrant activation of the mammalian (mechanistic) target of rapamycin (mTOR) signaling has been identified in multiple human tumors (*Hare and Harvey, 2017*; *Ortolani et al., 2015*) and is involved in resistance to endocrine therapy (*Johnston, 2015*; *Osborne and Schiff, 2011*). Everolimus, an analog of rapamycin, binds to the immunophilin FK Binding Protein-12 (FKBP-12) to generate a complex that inhibits the activation of mTOR, a key regulatory kinase (*Faivre et al., 2006*; *Law et al., 2006*; *Mita et al., 2003*). Everolimus in combination with the aromatase inhibitor exemestane improved

**eLife digest** Breast cancer is one of the most frequently diagnosed cancers globally, particularly among women. The most common type of breast cancer expresses a receptor for the hormone estrogen. Many treatments block the activity of estrogen and therefore slow or block the development and spread of this type of breast cancer.

For patients with advanced breast cancer, hormone-blocking treatments work best in combination with other drugs, including one called everolimus. However, in many patients the cancer cells become resistant to these therapies, leading to disease progression and decreased survival.

To explore treatment strategies that could enhance the effectiveness of existing therapies for breast cancer, Farmaki et al. studied how cancer cells which had become resistant to everolimus responded when treated with an experimental drug called ONC201/TIC10.

A combination of everolimus and ONC201/TIC10 inhibited growth of resistant cancer cells that had been grown in a three-dimensional arrangement to mimic human tumors. Moreover, the drug combination effectively targeted breast cancer cells collected from patients whose cancer had progressed while being treated with everolimus, suggesting that ONC201/TIC10 could be relevant in a clinical setting. Finally, molecular and biochemical experiments revealed that the drug ONC201/TIC10 works by disrupting the pathways that everolimus-resistant cancer cells use to generate the energy required to grow and proliferate.

Taken together these findings suggest that ONC201/TIC10 may provide an effective add-on therapy for patients with certain types of advanced breast cancer that are no longer responding to everolimus. Before this becomes a reality for patients, however, there will have to be more experimental testing of ONC201/TIC10 to determine optimal dosing and timing strategy for future clinical trials.

progression-free survival in patients with advanced BC (*Baselga et al., 2012*). Based on these clinical studies the combination was approved in 2012 as a second-line therapy for recurrent or mBC (National Comprehensive Cancer Network, NCCN guidelines) (*Baselga et al., 2012*). Despite its efficacy, a large subset of patients develops progression and resistance to this combination, highlighting the need for more effective therapeutic strategies for advanced BC.

Studies on everolimus resistance in BC revealed survival mechanisms including activation of Protein kinase B family (AKT) (*Carew et al., 2011*; *Chen et al., 2013*) and Mitogen-activated protein kinase (MAPK) signaling (*Campone et al., 2021*; *Carew et al., 2011*; *Kimura et al., 2018*; *Mendoza et al., 2011*), upregulation of MYC (*Bihani et al., 2015*; *Lui et al., 2016*), increased oxidative stress (*Neklesa and Davis, 2008*), metabolic reprogramming (*Pusapati et al., 2016*), increased expression of anti-apoptotic molecules such as survivin (*Taglieri et al., 2017*), and activation of epithelial and mesenchymal transition (*Holder et al., 2015*).

ONC201/TIC10 belongs to the imipridone class of inhibitors and is currently being evaluated in clinical trials for solid tumors including breast and endometrial cancer, gliomas, and hematological malignancies (*Prabhu et al., 2020*). The mechanism of ONC201/TIC10 action involves binding and activation of the mitochondrial protease caseinolytic protease P (ClpP) (*Greer et al., 2022*; *Jacques et al., 2020*; *Graves et al., 2019*; *Ishizawa et al., 2019*) and inhibition of the G-protein-coupled receptor (GPCR) dopamine receptor D2 (DRD2) (*Kline et al., 2018*). ONC201/TIC10 has been shown to synergize with everolimus in prostate cancer (*Lev et al., 2018*). Furthermore, colorectal cancer cells were more sensitive to the combination of ONC201/TIC10 with mTOR inhibitor AZD-8055 (*Jin et al., 2016*). While the effectiveness of the combination of everolimus with ONC201/TIC10 has not been studied in BC, upregulation of mitochondrial ClpP that is a target of ONC201/TIC10 has been reported both in BC cells and tissues (*Cormio et al., 2021*; *Luo et al., 2020*). In addition, ONC201/TIC10 inhibits mitochondrial oxidative phosphorylation in BC cells, causing depletion of cellular ATP (*Dwucet et al., 2021*; *Greer et al., 2018*; *Ishida et al., 2018*; *Pruss et al., 2020*). Increased dependency on oxidative phosphorylation (OXPHOS) has been shown in chemotherapy resistant triple negative BC (*Evans et al., 2021*). Therefore, targeting ClpP and mitochondrial metabolism could constitute a valid approach for increasing the efficacy of everolimus.

The aim of our study was to explore treatment strategies that can enhance the anti-tumor activity of standard of care therapy by targeting common resistant states specific to ER+ mBC. For our study, we developed in vitro models of acquired everolimus resistance for paired ER+ BC cell lines that are either sensitive or resistant to everolimus, and viable primary patient ER+ BC cells with known response to everolimus therapy. We used these models to identify therapies effective in everolimus resistant cancer cells and found that the small molecule ONC201/TIC10 can inhibit the proliferation of resistant cells by disrupting mitochondrial function and metabolism. RNA-sequencing analysis in ER+ mBC tumors non-responsive to everolimus and mitochondrial functional assays in resistant cell lines demonstrated increased dependency on mitochondrial respiration, further supporting the sensitivity to ONC201/TIC10. Thus, we demonstrate that ONC201/TIC10 enhances the durability of everolimus in resistant ER+ BC cell lines as well as primary cells from patients progressing on everolimus.

## Results

### ONC201/TIC10 inhibits the proliferation of BC cell lines sensitive and resistant to everolimus

Everolimus resistant cell lines were generated by long-term culture of parental cell lines in the continuous presence of 100 nM everolimus for MCF7 and T47D or 50 nM everolimus for CAMA-1 for 6–12 months (*McDermott et al., 2014*). Parental untreated cell lines were maintained in culture for the same time as the everolimus-treated cell lines as paired everolimus sensitive cell lines. To quantify the extent of resistance, we performed dose–response assays with increasing concentrations of everolimus in sensitive and everolimus resistant cells. Resistant cells showed a statistically significant reduced response to everolimus compared to sensitive cells ($p < 0.05$ for all paired cell lines, *Figure 1A*).

Next, we assessed the anti-proliferative efficacy of ONC201/TIC10 in the everolimus sensitive and resistant cell lines using both two-dimensional (2D) and three-dimensional (3D) assays. Sensitive and resistant cells were labeled with lentivirus to express a fluorescent protein for monitoring each population's growth when cultured (Venus and mCherry fluorescence for the sensitive and resistant cells, respectively). For the 2D assays, everolimus sensitive and resistant cells were treated with everolimus, ONC201/TIC10, or combination for 4 days. For the 3D assays, everolimus sensitive or resistant spheroids were cultured in the presence of drug treatments for 18 days with media and drug replacement every 3–4 days. Images and fluorescence signal intensity measurements were also captured on days of media and drug replacement to monitor growth inhibition over time.

In the 2D assays, ONC201/TIC10 single agent inhibited the proliferation of CAMA-1, MCF7, and T47D sensitive and resistant cells in a dose-dependent manner with IC50 values ranging from 1.43 to 1.90 µM (*Figure 1—figure supplement 1*). ONC201/TIC10 single agent treatment resulted in significant growth inhibition compared to control and everolimus single agent in all the resistant cell lines and in the CAMA-1 and MCF7 sensitive cells (*Figure 1B*). Combination of ONC201/TIC10 with everolimus significantly decreased cell proliferation compared to ONC201/TIC10 single agent, in all the resistant cell lines and in the MCF7 and T47D sensitive cells, by approximately 3% decrease for CAMA-1 sensitive, 5.5% for CAMA-1 resistant, 8% for MCF7 sensitive, 5% for MCF7 resistant, 8.8% for T74D sensitive, and 4.7% for T47D resistant (*Figure 1B*). Analysis of the drug interactions using the Bliss Interaction Index showed additive effect in all sensitive cell lines and T47D resistant cells and synergistic effect in the CAMA-1 and MCF7 resistant cells (*Figure 1C*).

In the 3D assays, ONC201/TIC10 single agent treatment resulted in significant growth inhibition compared to control in the all the resistant cell lines and in the T47D sensitive cells as indicated by the fluorescence intensity (*Figure 2A, B*). Even though everolimus single agent had increased inhibitory effect compared to ONC201/TIC10 single agent, the combination significantly inhibited spheroid growth in sensitive and resistant cell lines compared to both everolimus and ONC201/TIC10 single agent (*Figure 2A, B*). Drug interactions analysis showed moderately better than additive effect in the CAMA-1 resistant cells as well as in T47D sensitive and resistant cells (*Figure 2C*).

### Combination therapy of ONC201/TIC10 and everolimus inhibits the growth of primary patient-derived ER+ BC cell spheroids

To further validate the effect of the combination of ONC201/TIC10 with everolimus in cells from patients' tumors, we performed 3D assays with primary cells from ascites or pleural effusion from

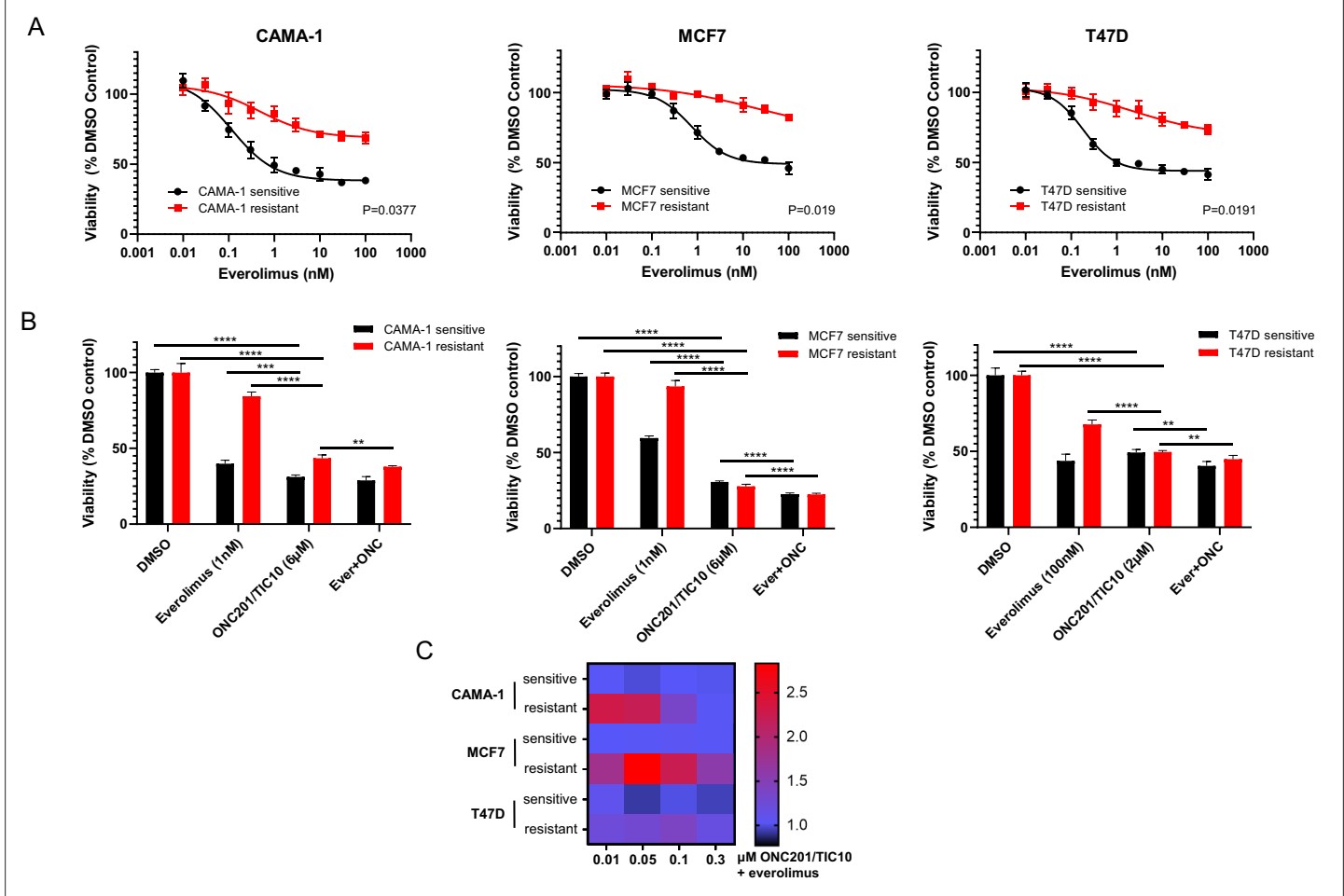

**Figure 1.** ONC201/TIC10 inhibits the proliferation of everolimus sensitive and resistant cells in 2D. (**A**) Dose–response curves of CAMA-1, MCF7, and T47D everolimus sensitive and resistant cells under everolimus treatment. Cells were treated with increasing concentration of everolimus for 4 days and viability was measured using CellTiterGlo Chemiluminescent kit. (**B**) Cell viability after 4 days treatment with Dimethyl sulfoxide (DMSO), everolimus, ONC201/TIC10, or combination at the indicated concentrations. Data represent % viable cells compared with DMSO control treatment for each cell line and are shown as average of four replicates ± standard deviation (SD). **p < 0.01, ***p < 0.001, ****p < 0.0001. (**C**) Analysis of ONC201/TIC10 and everolimus interactions in 2D. Cells were treated with 1 nM everolimus (CAMA-1 and T47D) or 100 nM everolimus (MCF7), ONC201/TIC10, or combination at the indicated concentrations for 4 days in 2D and viability was measured using CellTiterGlo Chemiluminescent kit. The average Bliss Interaction Index was calculated and plotted as a heatmap in which red represents synergy and blue represents additivity.

The online version of this article includes the following figure supplement(s) for figure 1:

**Figure supplement 1.** ONC201/TIC10 inhibits the proliferation of everolimus sensitive and resistant cells in 2D.

patients' tumor samples. We used short-term experiments, compared to sensitive and resistant cell lines, since primary cells were more sensitive to the drug treatments. Patient samples were selected based on the treatment history, specifically refractory tumors from patients who had received hormonal therapy followed by several lines of treatment including everolimus and progressed (*Table 1*). Spheroids of patient-derived BC cells illustrate more accurately the characteristics of tumors in vivo and provide a more suitable model for the assessment of drug treatments compared with the 2D cultures (*Imamura et al., 2015*; *Langhans, 2018*). Spheroids from five different patients were treated with ONC201/TIC10, everolimus, or combination for 4 days. The effect of drug treatment on growth inhibition was assessed by measurement of cell proliferation (*Figure 3A*). Our results indicate that combination therapy of ONC201/TIC10 with everolimus inhibits spheroid growth compared to everolimus single agent or ONC201/TIC10 single agent (ONC201/TIC10 single agent vs. combination 12.5% cell proliferation decrease for P1, 11% for P2, 4% for P3, 0.2% for P4, and 10.5% for P5) (*Figure 3A*). Importantly, we compared the response of patient's cells during everolimus treatment

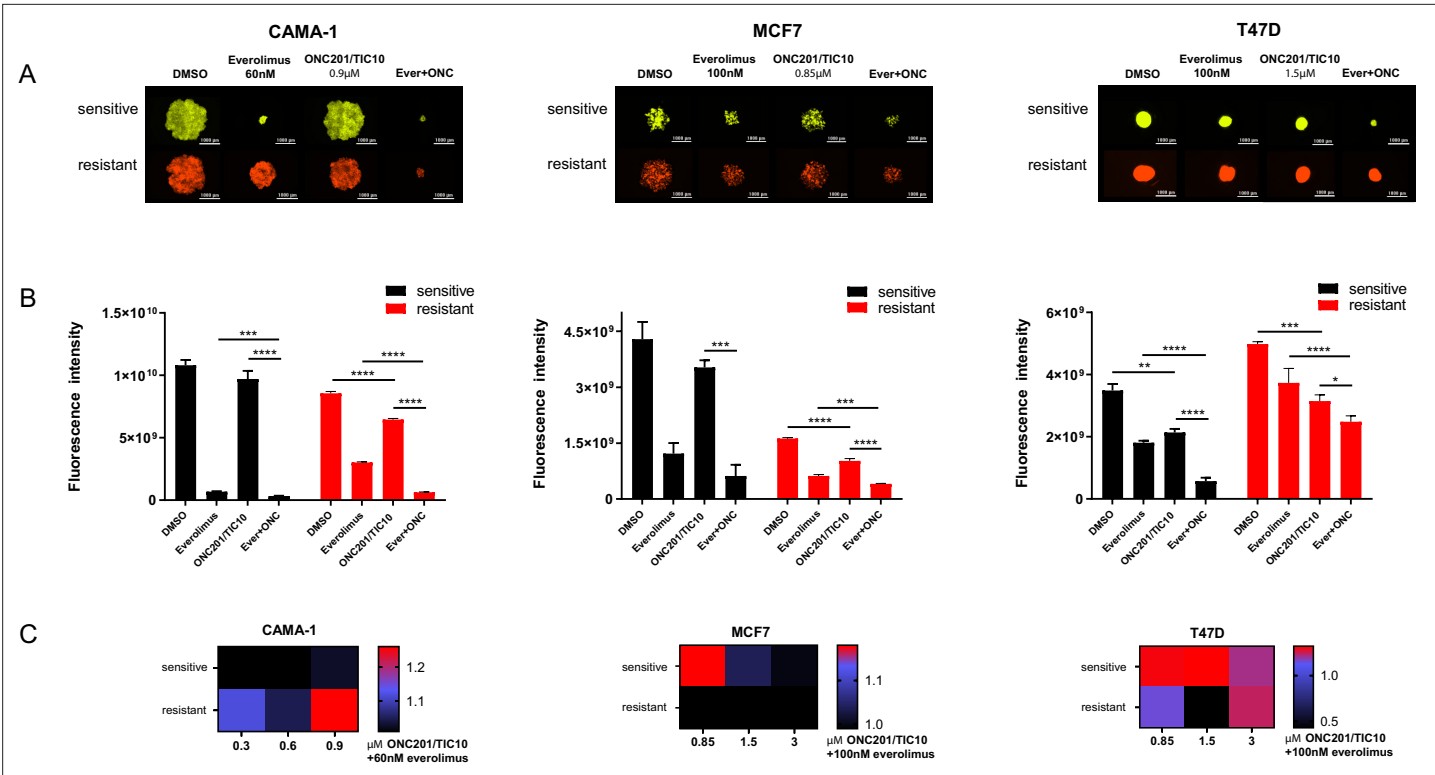

**Figure 2.** Combination of ONC201/TIC10 and everolimus inhibits spheroid growth in 3D. (**A**) Representative images of spheroid growth of sensitive (Venus, green) or resistant (mCherry, red) cells cultured in everolimus, ONC201/TIC10, or combination treated media at the indicated concentrations for up to 18 days. (**B**) Fluorescence intensity of sensitive and resistant cells under various treatment conditions. Data are represented as average of three replicates ± standard deviation (SD). *p < 0.05, **p < 0.01, ***p < 0.001, ****p < 0.0001. (**C**) Analysis of ONC201/TIC10 and everolimus interactions in 3D. Spheroids were cultured in the presence of drug treatments for 18 days and fluorescence intensity measurements were captured. The average Bliss Interaction Index was calculated and plotted as a heatmap in which red represents synergy and blue represents additivity.

and after everolimus resistance, based on the availability of samples for P2 that were collected over the course of treatment. We used these two different tumor samples, from the same patient, during everolimus treatment or post-everolimus and during chemotherapy treatment (*Figure 3B*). ONC201/TIC10 single agent or combination of ONC201/TIC10 with everolimus exhibited greater growth inhibition in the post-everolimus treatment cells (*Figure 3B*, *Figure 3—figure supplement 1*) as shown by the significant reduction in proliferation of these cells compared to the cells acquired during everolimus treatment (7.2% cell decrease in cell proliferation during everolimus vs. post-everolimus with ONC201/TIC10 and 9.2% cell decrease in cell proliferation during everolimus vs. post-everolimus with the combination) (*Figure 3B*) and decreased spheroid size (*Figure 3—figure supplement 1*). These results highlight the efficacy of ONC201/TIC10 on the patient-derived resistant cells.

**Table 1.** Treatment history of patients included in the study.

| | Patient diagnosis | Sample | Therapy lines | Drugs | Therapy lines post-everolimus |
|---|---|---|---|---|---|
| P1 | Metastatic ER+/HER2− | Ascites | 6 | 12 | 2 |
| P2 | Metastatic ER+/PR+/HER2− | Ascites | 7 | 9 | 2 |
| P3 | Metastatic ER+/PR+/HER2− | Pleural effusion | 7 | 9 | 1 |
| P4 | Metastatic ER+/PR+/HER2− | Ascites | 10 | 14 | 5 |
| P5 | Metastatic ER+/PR+/HER2− | Ascites | 5 | 6 | 1 |

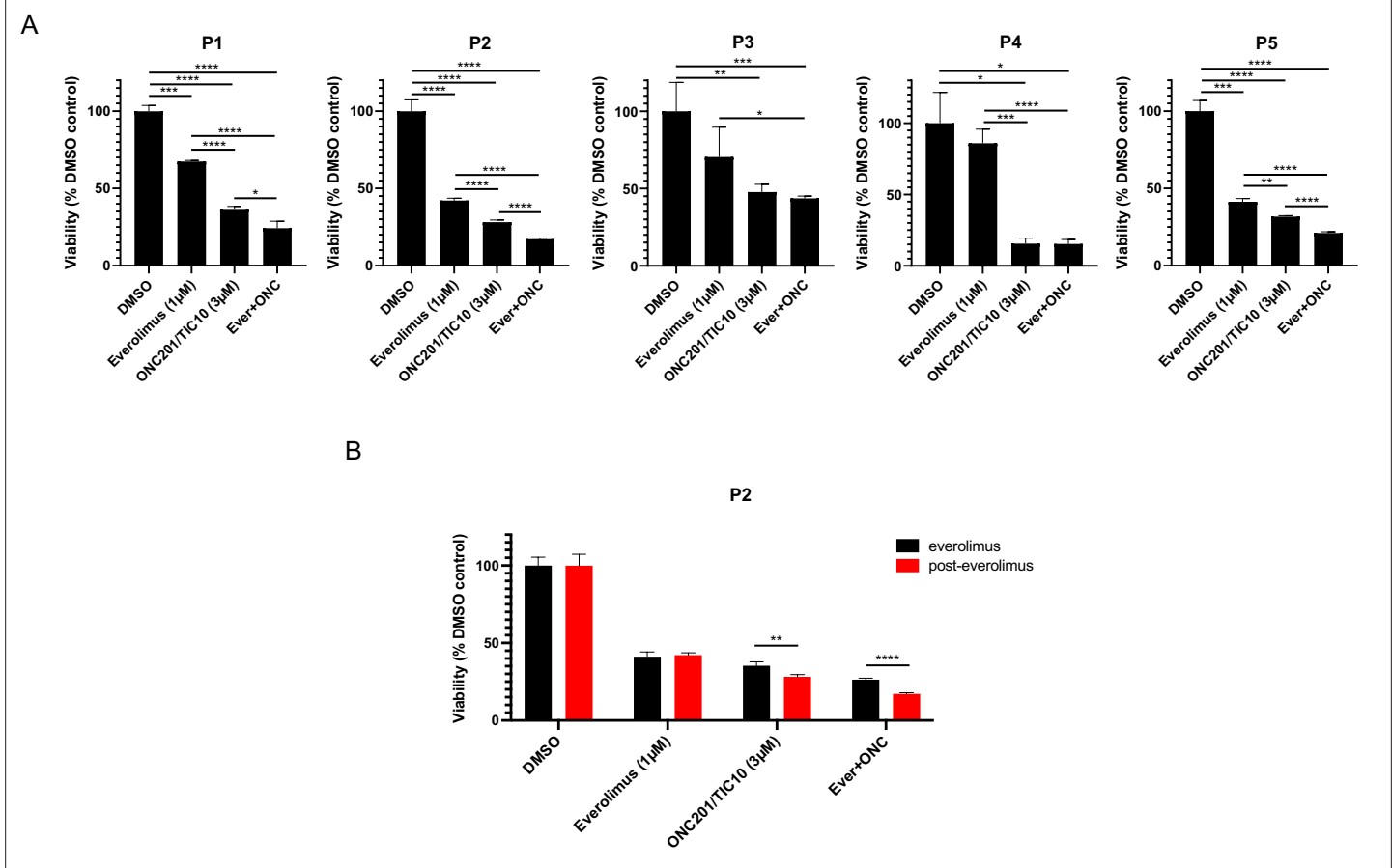

**Figure 3.** Combination therapy of ONC201/TIC10 and everolimus inhibits the growth of primary patient-derived cell spheroids. (**A**) 3D cultures of primary patient-derived ER+ BC cells from ascites or pleural effusion treated with everolimus, ONC201/TIC10, or combination at the indicated concentrations for 4 days. Cell viability was measured using CellTiterGlo Chemiluminescent kit. (**B**) 3D cultures of primary patient-derived ER+ BC cells, while on everolimus treatment or post-everolimus treatment. Spheroids were treated with everolimus, ONC201/TIC10, or combination at the indicated concentrations for 4 days. Data represent % viable cells compared with DMSO control treatment and are shown as average of three replicates ± standard deviation (SD). *p < 0.05, **p < 0.01, ***p < 0.001, ****p < 0.0001.

The online version of this article includes the following figure supplement(s) for figure 3:

**Figure supplement 1.** Combination therapy of ONC201/TIC10 and everolimus inhibits the growth of primary patient-derived cell spheroids.

## ONC201/TIC10 causes loss of mitochondrial proteins and activation of stress response in everolimus sensitive and resistant cells

To identify the mechanism of ONC201/TIC10 sensitivity in the sensitive and resistant cells, we investigated the effects of the treatments on downstream signaling pathways, including mitochondrial pathways and stress pathways that have been previously described for BC cells under ONC201/TIC10 treatment (*Greer et al., 2018*; *Ralff et al., 2017*; *Yuan et al., 2017*). Western blot analysis showed that ONC201/TIC10 suppresses the expression of the mitochondrial proteins TFAM (mitochondrial transcription factor A) and TUFM (translation elongation factor Tu) in all sensitive and resistant cell lines and specifically in a dose-dependent manner in CAMA-1 and T47D sensitive and resistant cells (*Figure 4A*). TFAM and TUFM are important regulators of mitochondrial functions and are associated with the ONC201/TIC10-induced mitochondrial disruption (*Graves et al., 2019*; *Greer et al., 2018*).

We further assessed the effect of ONC201/TIC10 on the oxidative phosphorylation proteins since ONC201/TIC10 toxicity has been linked to impaired OXPHOS in breast and brain tumors (*Dwucet et al., 2021*; *Greer et al., 2018*; *Ishida et al., 2018*; *Pruss et al., 2020*). ONC201/TIC10 suppresses the expression of the respiratory chain complexes in sensitive and resistant cells, specifically Complex I subunit NDUFB8, Complex II subunit 30 kDa (CAMA-1 and MCF7), Complex III subunit Core 2, and ATP synthase subunit alpha (*Figure 4A*).

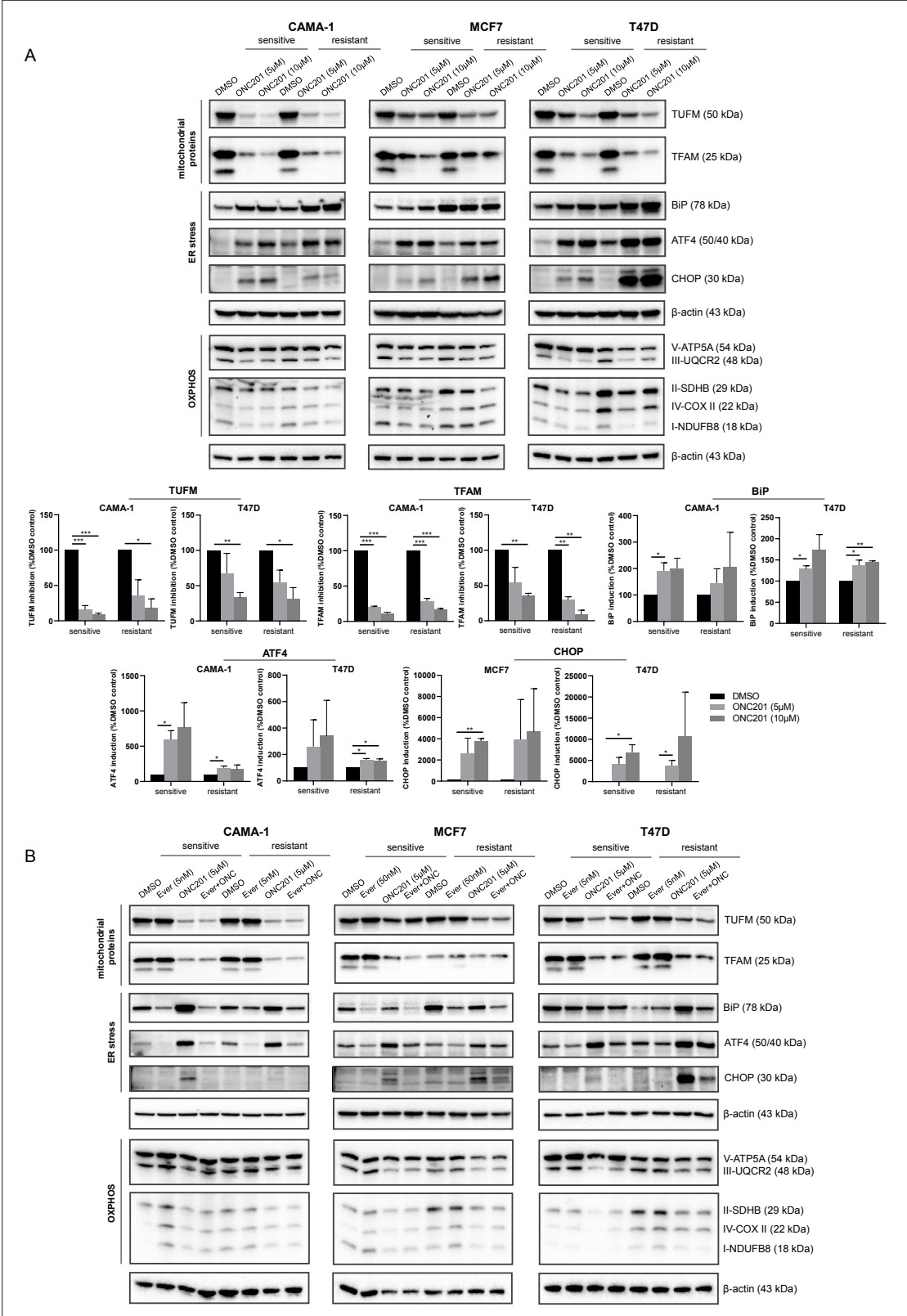

**Figure 4.** ONC201/TIC10 causes loss of mitochondrial proteins and activation of stress response in everolimus sensitive and resistant cells. (**A**) CAMA-1, MCF7, T47D everolimus sensitive and resistant cells were treated for 24 hr with ONC201/TIC10 at the indicated concentrations, and cell lysates were immunoblotted for TUFM, TFAM, BiP, ATF4, CHOP, and OXPHOS complexes (Complex I subunit NDUFB8, Complex II subunit 30 kDa, Complex III subunit Core 2, Complex IV subunit II, and ATP synthase subunit alpha), and β-actin. Quantitation of TUFM, TFAM, BiP, ATF4, and CHOP using ImageJ

*Figure 4 continued*

analysis. Protein expression levels were normalized to β-actin and are shown as average of two replicates ± standard deviation (SD). *p < 0.05, **p < 0.01, ***p < 0.001. (**B**) CAMA-1, MCF7, T47D everolimus sensitive and resistant cells were treated for 24 hr with everolimus, ONC201/TIC10, and combination at the indicated concentrations, and cell lysates were immunoblotted for the same proteins as above.

The online version of this article includes the following source data and figure supplement(s) for figure 4:

**Source data 1.** Original files of the full blot images for *Figure 4A*.

**Source data 2.** Original files of the full blot images for *Figure 4B*.

**Source data 3.** Figures with uncropped blot images for *Figure 4A, B*.

**Figure supplement 1.** ONC201/TIC10 mechanism in everolimus sensitive and resistant cells is TRAIL independent.

**Figure supplement 1—source data 1.** Original files of the full blot images for *Figure 4—figure supplement 1*.

**Figure supplement 1—source data 2.** Figures with uncropped blot images for *Figure 4—figure supplement 1*.

Next, we evaluated in our model the activation of the Integrated Stress Response (ISR), that has been reported to follow mitochondrial dysfunction in breast (*Greer et al., 2018*; *Ralff et al., 2017*; *Yuan et al., 2017*) and other malignancies (*Al Madhoun et al., 2021*; *Fan et al., 2022*; *Lev et al., 2018*). ONC201/TIC10 induces the expression of the chaperone BiP (HP70) in all sensitive cell lines, and specifically in a dose-dependent manner in CAMA-1 and T47D sensitive and resistant cells (*Figure 4A*). Furthermore, ONC201/TIC10 induces the expression of the transcription factor ATF4 in all sensitive and resistant cell lines, with a dose-dependent effect in CAMA-1 and T47D sensitive cells (*Figure 4A*). The transcription factor CHOP is induced after treatment in all sensitive and resistant cell lines, with a dose-dependent effect in MCF7 and T47D sensitive and resistant cells (*Figure 4A*).

Combination treatment of ONC201/TIC10 with everolimus had the same effect on the suppression of the mitochondrial proteins TFAM and TUFM and the OXPHOS complexes as the single agent (*Figure 4B*). Everolimus single agent did not affect the mitochondrial protein levels, as expected (*Figure 4B*). Activation of stress response was not observed with the combination in the sensitive cells (*Figure 4B*).

Inhibition of AKT, ERK, and FoxO3a phosphorylation and activation of TRAIL that were previously described for ONC201/TIC10 (*Allen et al., 2013*; *Ralff et al., 2017*) were not observed in our model system (*Figure 4—figure supplement 1*). Furthermore, ONC201/TIC10 inhibits the phosphorylation of ribosomal protein S6 that is downstream of mTOR signaling and this inhibition is increased in the resistant cell lines (*Figure 4—figure supplement 1*).

To corroborate the mechanism of ONC201/TIC10 action observed in our experiments, we analyzed time course gene expression data of the MDA-MB-231 BC cell line treated with ONC201/TIC10 (*Greer et al., 2018*). Using a generalized linear model, we investigated the changes in pathway activity levels of the cell line measured at 0, 3, 6, 12, and 24 hr after treatment. Over the time course, ONC201/TIC10 treatment significantly reduced cell cycle pathway activity in the cell line ($p = 1.8 \times 10^{-6}$) (*Figure 5A*). Concurrently, we found a significant increase in the ATF4 target genes activated in response to ER stress ($p = 1.2 \times 10^{-4}$) (*Figure 5B*) and the UPR pathway activity ($p = 1.8 \times 10^{-4}$) (*Figure 5C*). In contrast, neither of the ERK ($p = 0.55$) (*Figure 5D*) or AKT ($p = 0.61$) (*Figure 5E*) signaling pathways showed any change in activity over time.

## ONC201/TIC10 inhibits mitochondrial respiration in everolimus sensitive and resistant cells

Based on the findings for the mitochondrial pathways, we then assessed the mitochondrial respiratory capacity of sensitive and resistant cell lines using extracellular flux analysis. Comparison of the mitochondrial respiration between the sensitive and resistant cells indicates that CAMA-1 resistant cells have lower basal and maximal respiratory capacity compared to the sensitive cells while MCF7 and T47D resistant cells have significantly increased baseline, maximal and ATP-linked respiration compared to the sensitive cells suggesting increased dependency on OXPHOS for ATP production (*Figure 6—figure supplement 1*). Furthermore, all resistant lines showed significantly increased baseline and stressed extracellular acidification rate (ECAR) (*Figure 6—figure supplement 1*). Treatment with ONC201/TIC10 resulted in a significant decrease in oxidative consumption rates (OCR) compared to control and everolimus single agent, in both sensitive and resistant cell lines at baseline, maximal and ATP-linked respiration (*Figure 6*, *Figure 6—figure supplement 2*). In addition, combination

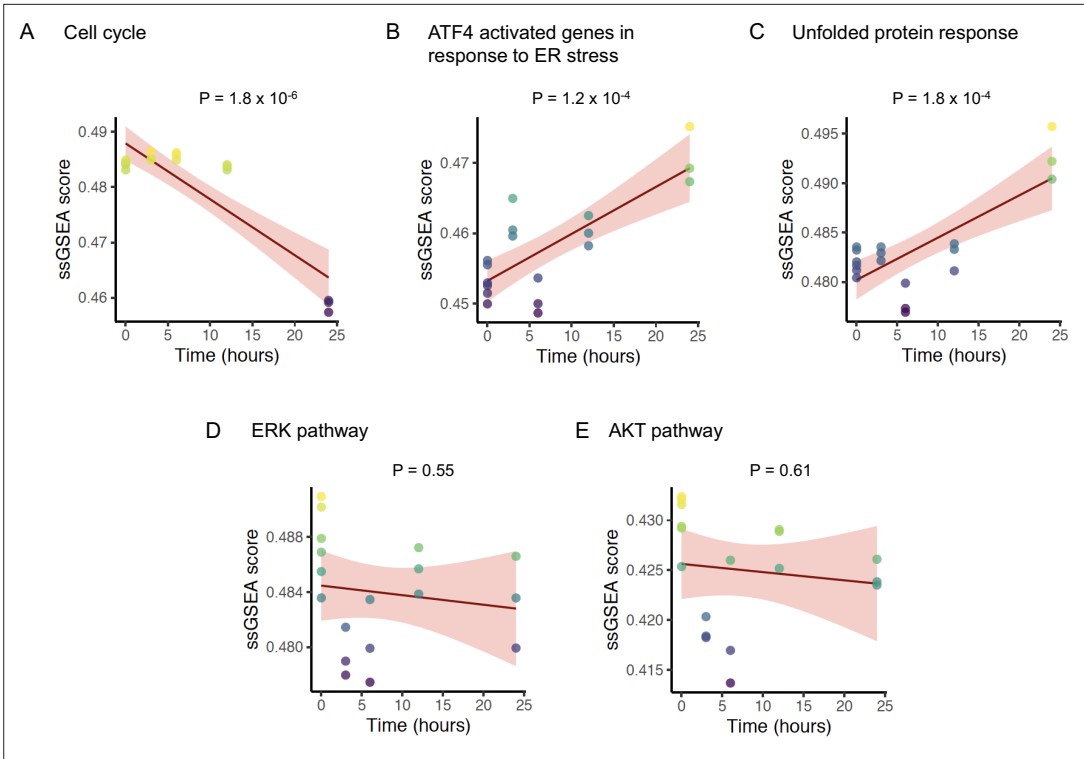

**Figure 5.** Change in pathway activity over time in response to ONC201/TIC10. Scatter plots displaying the enrichment scores (*Y*-axis) of (**A**) REACTOME cell cycle signature, (**B**) REACTOME ATF4 activated genes in response to endoplasmic reticulum stress signature, (**C**) REACTOME unfolded protein response UPR signature, (**D**) BIOCARTA ERK pathway signature, and (**E**) BIOCARTA AKT pathway signature over time (*X*-axis). The solid lines and shaded area indicate linear fit and 95% confidence intervals, respectively, with p-value of the fit indicated above each plot. The analysis includes data from six replicates at time 0 hr, and three replicates each at time 3, 6, 12, and 24 hr for a total *n* = 18.

treatment of ONC201/TIC10 with everolimus further reduced mitochondrial respiratory capacities in all sensitive and resistant cells, except CAMA-1 cells that ONC201/TIC10 alone has similar effect on OCR as the combination (*Figure 6*, *Figure 6—figure supplement 2*).

## Tumors non-responsive to everolimus sustain mitochondrial oxidative phosphorylation activity

We compared the changes in pathway phenotypes of mTOR inhibitor sensitive and resistant tumors using gene expression data from a neoadjuvant clinical trial of ER+ BC patients receiving everolimus (*Sabine et al., 2010*). We examined the changes in 21 patients by comparing single sample gene set enrichment scores of pathway signatures from the RNA-sequencing data from this trial taken before and after 11–14 days of treatment with everolimus. Patients classified as everolimus responders based on decrease in Ki67 staining post-treatment displayed significant decrease in cell cycle signature (p = $4.3 \times 10^{-3}$), while non-responders did not show a significant change (p = 0.2) (*Figure 7A*). This result confirmed that everolimus failed to control the growth and proliferation of resistant tumors in an independent patient dataset. In the responder group, we found that everolimus treatment led to a significant decrease in the mitochondrial oxidative phosphorylation (p = $7.6 \times 10^{-3}$), while there was no significant difference in the non-responders' group (p = 0.83) (*Figure 7B*).

## Discussion

Therapies targeting signaling pathways such as the CDK4/6 and the PI3K/AKT/mTOR pathway that have been identified among the pathways involved in endocrine therapy resistance (*Johnston, 2015*) have provided a significant clinical benefit in ER+ BC. The BOLERO-2 study showed that addition of

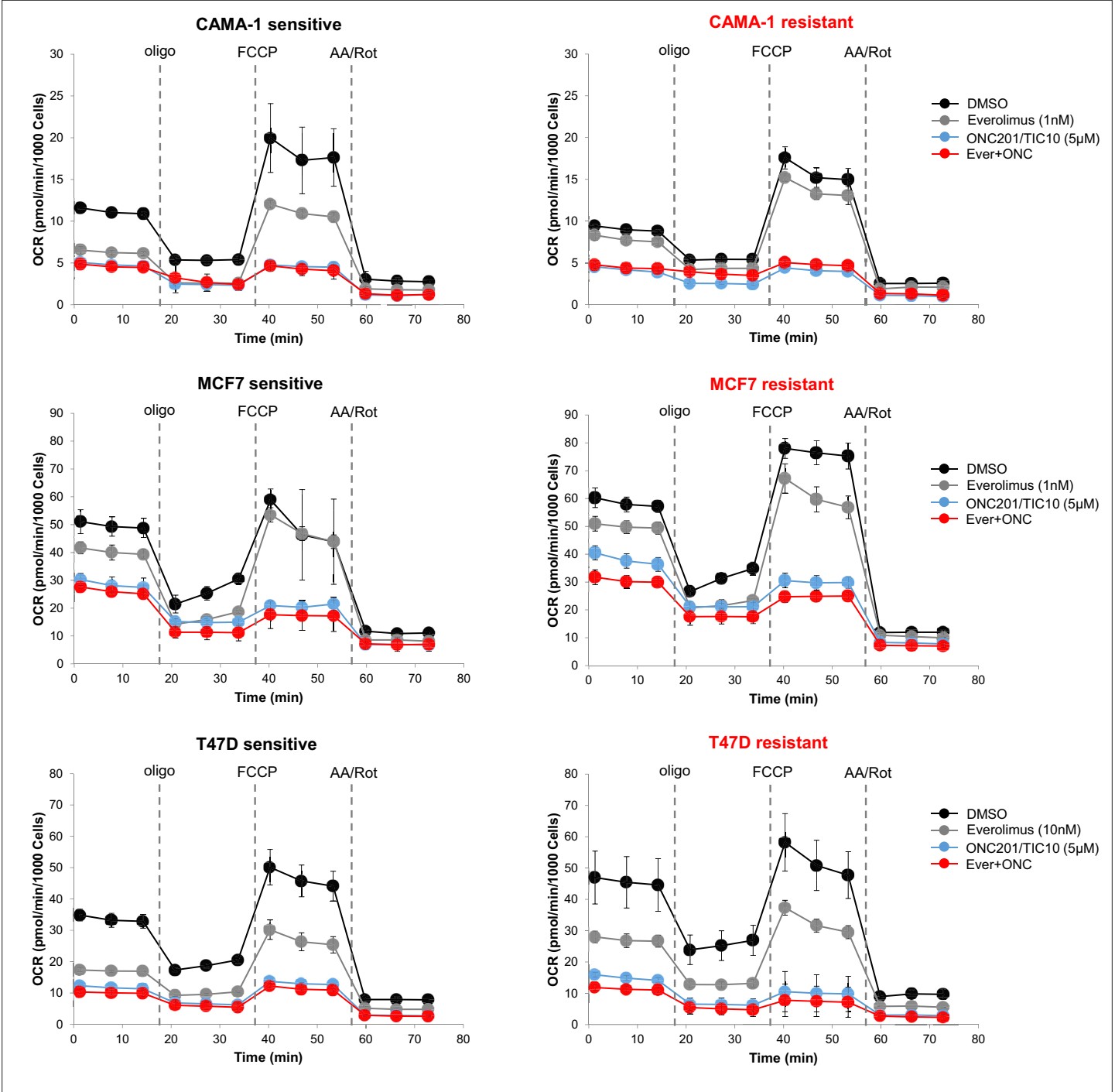

**Figure 6.** ONC201/TIC10 inhibits mitochondrial respiration in everolimus sensitive and resistant cells. Cells were treated for 18 hr with indicated concentrations of everolimus, ONC201/TIC10, and combination. Mitochondrial respiration was measured using Seahorse XF Cell Mito Stress assay and oxygen consumption rates (OCR) are shown. Values were normalized to cell number generated from fluorescence intensity measurements and are represented as average of three replicates.

The online version of this article includes the following figure supplement(s) for figure 6:

**Figure supplement 1.** Mitochondrial respiration in everolimus sensitive and resistant cells.

**Figure supplement 2.** ONC201/TIC10 inhibits mitochondrial respiration in everolimus sensitive and resistant cells.

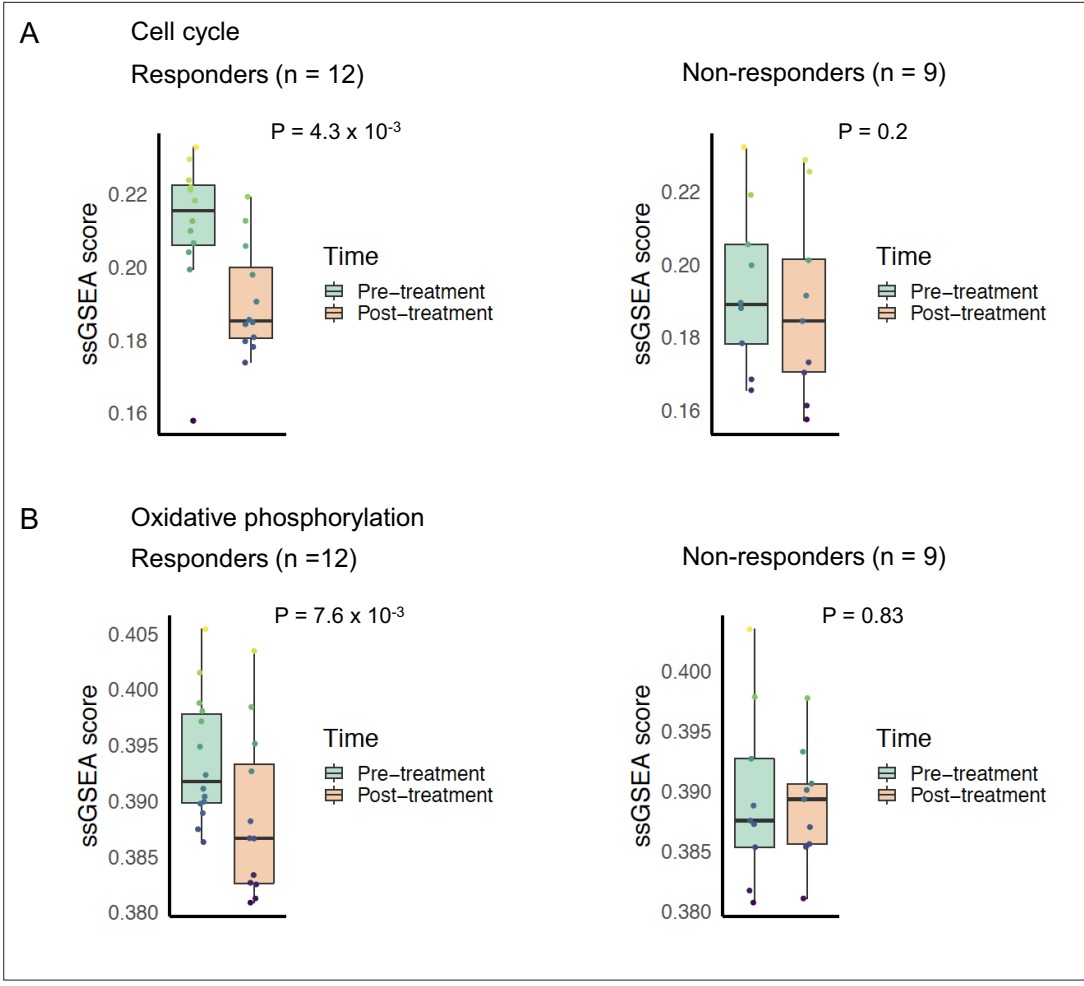

**Figure 7.** Cell cycle and oxidative phosphorylation pathway activity and neoadjuvant everolimus response. Box plots comparing cell cycle pathway activity indicated by (**A**) REACTOME cell cycle signature enrichment scores and (**B**) Kyoto Encyclopedia of Genes and Genomes (KEGG) oxidative phosphorylation signature enrichment scores between pre- and post-treatment samples. The left panel shows patients that were classified as responders to everolimus, while the right panel shows patients that were classified as non-responders. Colored boxes indicate interquartile range, horizontal bars indicate median, and the whiskers indicate first and third quartiles. p-values from paired two-tailed *t*-test comparing pre-treatment vs. post-treatment scores are indicated above the plots.

everolimus to exemestane significantly improves progression-free survival, in post-menopausal ER+ BCs from 2.8 months on exemestane alone to 6.9 months on everolimus plus exemestane (*Baselga et al., 2012*). Despite this progress, resistance to targeted therapies still emerges in a substantial proportion of patients. Therefore, it is important to explore strategies to increase response to this therapy.

In the present study, we explored treatment strategies to improve targeted therapy response and survival by combining drugs targeting resistance mechanisms. Specifically, we investigated if combination of everolimus with ONC201/TIC10 is effective for resistant BC cells and could provide a promising treatment strategy. Previous studies showed the efficacy of ONC201/TIC10 in TNBC cells (*Greer et al., 2018*; *Yuan et al., 2017*). We focused on the effect of ONC201/TIC10 on advanced ER+ BC cells including everolimus resistant cells and patient-derived spheroids resistant to endocrine therapy and everolimus.

Our results show that non-responsive tumors continued to proliferate after everolimus treatment by utilizing oxidative phosphorylation pathway to fuel the growth of tumor cells and is consistent with the findings from the resistant cell lines, supporting that ONC201/TIC10 and modulation of mitochondrial function is effective in drug resistant cancer cells. Combination therapy with ONC201/

TIC10 and everolimus showed increased growth inhibition in everolimus resistant BC cell lines as well as resistant patient tumor cells in 2D and 3D studies. In the 2D culture setting, ONC201/TIC10 was effective both as single agent and in combination with everolimus with similar potency in sensitive and resistant cells. In the 3D setting however, ONC201/TIC10 in combination with everolimus had a robust effect on growth inhibition in both resistant and sensitive cells, compared to ONC201/TIC10 single agent that had a minor effect. Differences in drug potency between the 2D and 3D cultures have been previously reported for BC cells and are associated with lower proliferation, increased apoptosis, microenvironmental conditions like hypoxia and reduced nutrients and altered signaling in the 3D setting (*Burdett et al., 2010*; *Hamilton, 1998*; *Lovitt et al., 2015*). These differences highlight the importance of utilizing more than one experimental model based on the hypotheses that are being addressed and given the limitations of each. Consistent with the resistant cell lines, the combination therapy had increased anti-proliferative activity in the resistant patient-derived spheroids. Collectively, these results recapitulate the increased sensitivity to the combination therapy and further support clinical relevance.

Mechanistically, ONC201/TIC10 has been shown to directly activate ClpP, through various genetic and biochemical studies (*Greer et al., 2022*; *Jacques et al., 2020*; *Graves et al., 2019*; *Ishizawa et al., 2019*). In BC, silencing of ClpP using siRNA or CRISPR/Cas9 System confirmed the ClpP-dependent effects of ONC201/TIC10 (*Graves et al., 2019*; *Greer et al., 2022*). Based on these established findings, we detail the specific pathway components driving the therapeutic response of ONC201/TIC10 in everolimus sensitive and resistant cells. We show that ONC201/TIC10 causes mitochondrial dysfunction, which is consistent with previous findings for BC cells, including suppression of mitochondrial proteins, decreased OXPHOS and activation of stress response (*Greer et al., 2018*). We observed this effect not only in the sensitive cells but also in everolimus resistant cells, which supports the similarity in potency of ONC201/TIC10 single agent in everolimus sensitive and resistant cells in 2D assays. Importantly, data from RNA-sequencing in everolimus resistant patient tumors and from mitochondrial functional assays in resistant cell lines demonstrated increased dependency on mitochondrial respiration supporting the sensitivity to ONC201/TIC10. Therefore, ONC201/TIC10 could be a suitable candidate for the treatment of everolimus resistant tumors that depend on mitochondrial oxidative phosphorylation activity for sustained growth and proliferation.

Combination of ONC201/TIC10 with everolimus maintained the same level of mitochondrial and OXPHOS protein suppression as ONC201/TIC10 single agent. However, the activation of ISR after combination treatment was maintained only in the everolimus resistant cells and not the sensitive. This result suggests that in the sensitive cells, everolimus alleviates endoplasmic reticulum stress through the activation of different cell death mechanisms. Together these findings support the increased sensitivity of the combination in everolimus resistant cells and suggest as a mechanism of anti-proliferative activity the disruption of mitochondria and activation of stress response. Furthermore, western blot analysis and gene expression data confirm that the observed activity of ONC201/TIC10 is via induction of mitochondrial stress rather than ERK/AKT inactivation. Increased gene expression and protein levels of pro-apoptotic transcription factors CHOP and ATF4 have been reported for ONC201/TIC10 treatment (*Greer et al., 2018*; *Ishizawa et al., 2016*; *Kline et al., 2016*). Indeed, CHOP has been shown to induce apoptosis through the regulation of different anti- and pro-apoptotic genes, including genes encoding the Bcl-2 family proteins, GADD34, endoplasmic reticulum oxidoreductin 1 (ERO1α), Tribbles-related protein 3 (TRB3), and DOC (*Hu et al., 2018*; *Szegezdi et al., 2006*). In addition, CHOP can downregulate the expressions of Bcl-2, Bcl-xL, and Mcl-1, and upregulate the expression of BIM, causing increased BAK and BAX expression. Bcl-2, Bcl-xL, and Mcl-1 have been shown to be downregulated by imipridones and ONC201/TIC10 (*Al Madhoun et al., 2021*; *Rumman et al., 2021*; *Staley et al., 2021*).

ATF4 can promote apoptosis either through regulating CHOP or independent of CHOP (*Wortel et al., 2017*). ATF4 also downregulates the anti-apoptotic Bcl-2 protein and upregulates pro-apoptotic signaling through the proteins BIM, NOXA, and PUMA (*Szegezdi et al., 2006*; *Wortel et al., 2017*). ONC201/TIC10 caused apoptosis through ATF4 in lymphoma and leukemia cells and inhibited mTORC1 signaling through the upregulation of ATF4 and DDIT4 (*Ishizawa et al., 2016*; *Wang and Dougan, 2019*).

Our results show that ISR was not activated upon combination treatment with ONC201/TIC10 and everolimus in sensitive cells, suggesting that sensitivity to everolimus in these cells can result in the

activation of cell death mechanisms bypassing ISR. In line with this assumption, mTOR inhibitors are known to induce apoptosis through decreasing expression levels of various anti-apoptotic proteins, including Bcl-2, Bcl-xL, Mcl-1, and survivin (*Du et al., 2018*; *Mills et al., 2008*; *Wangpaichitr et al., 2008*). Other mechanisms of ONC201/TIC10 anti-proliferative activity in the everolimus resistant cells might include the involvement of c-Myc. High levels of c-Myc have been shown to be a predictive factor for growth inhibition and apoptosis by imipridones in glioblastoma (*Ishida et al., 2018*). In addition, the role of the Myc gene in promoting mTOR inhibitor resistance has been described in ER+ BC (*Bihani et al., 2015*).

In conclusion, we used preclinical and clinical models to characterize sensitivity to mTOR inhibition. Combining experimental and clinical patient-derived data with resistant cell line driven experiments, our study provides validated findings that are consistent across different contexts, thereby strengthening the potential for clinical applications of ONC201/TIC10. Collectively, our findings suggest that ONC201/TIC10 could be used as an add-on treatment after mTOR therapy progression. Based on the initial preclinical and clinical testing of ONC201/TIC10 that demonstrated benefit with a benign clinical profile, ONC201/TIC10 is being further evaluated as single agent or in combination with other cancer therapies for various tumor types (*Prabhu et al., 2020*). Combination of ONC201/TIC10 with everolimus has been tested in an in vivo prostate model and was well tolerated, with no additional toxicity (*Lev et al., 2018*). Further in vitro as well as in vivo testing of this combination would be necessary to determine the optimal dosing and timing strategy for future clinical trials.

# Materials and methods

## Key resources table

| Reagent type (species) or resource | Designation | Source or reference | Identifiers | Additional information |
|---|---|---|---|---|
| Cell line (*Homo-sapiens*) female | CAMA-1 Breast; Mammary gland: adenocarcinoma | ATCC | HTB-21 | |
| Cell line (*Homo-sapiens*) female | MCF7 Breast; Mammary gland: adenocarcinoma epithelial cell | ATCC | HTB-22 | |
| Cell line (*Homo-sapiens*) female | T47D Breast; Mammary gland: carcinoma ductal epithelial cell | ATCC | HTB-133 | |
| Cell line (*Homo-sapiens*) female | Primary, P1, P2, P3, P4, P5 | This paper. City of Hope IRB #07047 and #17334 | P1, P2, P3, P4, P5 | Pleural effusion or ascites samples from patients |
| Transfected construct (mammalian) | LeGO-V2 (Venus) | *Weber et al., 2008* Addgene #27340 | Addgene #27340; RRID:Addgene_27340 | Lentiviral construct to transfect and express Venus fluorescent protein |
| Transfected construct (mammalian) | LeGO-C2 (mCherry) | *Weber et al., 2008* Addgene #27339 | Addgene #27339; RRID:Addgene_27339 | Lentiviral construct to transfect and express mCherry fluorescent protein |
| Antibody | anti-BiP (C50B12) (rabbit monoclonal) | Cell Signaling Technology | Cat#3177; RRID:AB_2119845 | WB (1:1000) |
| Antibody | anti-pFoxO3a (Ser294) (rabbit polyclonal) | Cell Signaling Technology | Cat#5538; RRID:AB_10696878 | WB (1:1000) |
| Antibody | anti-FoxO3a (D19A7) (rabbit monoclonal) | Cell Signaling Technology | Cat#12829; RRID:AB_2636990 | WB (1:1000) |
| Antibody | anti-pERK1/2 Thr202/Tyr204 (197G2) (rabbit monoclonal) | Cell Signaling Technology | Cat#4377; RRID:AB_331775 | WB (1:1000) |
| Antibody | anti-ERK1/2 (137F5) (rabbit monoclonal) | Cell Signaling Technology | Cat#4695; RRID:AB_390779 | WB (1:1000) |
| Antibody | anti-pAKT Ser473 (D9E) (rabbit monoclonal) | Cell Signaling Technology | Cat#4060; RRID:AB_2315049 | WB (1:1000) |
| Antibody | anti-AKT (C67E7) (rabbit monoclonal) | Cell Signaling Technology | Cat#4691; RRID:AB_915783 | WB (1:1000) |

*Continued on next page*

*Continued*

| Reagent type (species) or resource | Designation | Source or reference | Identifiers | Additional information |
|---|---|---|---|---|
| Antibody | anti-pS6 Ser240/244 Ribosomal Protein (D68F8) (rabbit monoclonal) | Cell Signaling Technology | Cat#5364; RRID:AB_10694233 | WB (1:1000) |
| Antibody | anti-S6 Ribosomal Protein (54D2) (mouse monoclonal) | Cell Signaling Technology | Cat#2317; RRID:AB_2238583 | WB (1:1000) |
| Antibody | anti-β-actin (8H10D10) (mouse monoclonal) (HRP Conjugate) | Cell Signaling Technology | Cat#12262; RRID:AB_2566811 | WB (1:2000) |
| Antibody | anti-β-actin (13E5) (rabbit monoclonal) (HRP Conjugate) | Cell Signaling Technology | Cat#5125; RRID:AB_1903890 | WB (1:2000) |
| Antibody | anti-rabbit IgG, HRP-linked (goat polyclonal) | Cell Signaling Technology | Cat#7074; RRID:AB_2099233 | WB (1:2000) |
| Antibody | anti-mouse IgG, HRP-linked (horse polyclonal) | Cell Signaling Technology | Cat#7076; RRID:AB_330924 | WB (1:2000) |
| Antibody | anti-CREB-2/ATF-4 (B-3) (mouse monoclonal) | Santa Cruz Biotechnology | Cat#390063; RRID:AB_2810998 | WB (1:300) |
| Antibody | anti-mtTFA (TFAM) (C-9) (mouse monoclonal) | Santa Cruz Biotechnology | Cat#376672; RRID:AB_11150497 | WB (1:200) |
| Antibody | anti-TRAIL (55B709.3) (mouse monoclonal) | Thermo Fisher Scientific | Cat# MA1-41027; RRID:AB_1087999 | WB (1.5 µg/ml) |
| Antibody | anti-CHOP (rabbit polyclonal) | Proteintech | Cat#15204-1-AP; RRID:AB_2292610 | WB (1:1000) |
| Antibody | anti-TUFM (rabbit polyclonal) | Thermo Fisher Scientific | Cat#PA5-27511; RRID:AB_2544987 | WB (1:500) |
| Antibody | anti-Total OXPHOS (mouse monoclonal) | Abcam | Cat#ab110411; RRID:AB_2756818 | WB (1:1000) |
| Chemical compound, drug | Everolimus (RAD001) | Selleckchem | Cat#S1120 | Dissolved in DMSO |
| Chemical compound, drug | ONC201/TIC10 | Selleckchem | Cat#S7963 | Dissolved in DMSO |
| Commercial assay or kit | CellTiter-Glo Luminescent Cell Viability Assay | Promega | Cat#G7573 | |
| Commercial assay or kit | CellTiter-Glo 3D Cell Viability Assay | Promega | Cat#G9682 | |
| Commercial assay or kit | EasySep CD45 Depletion Kit II | StemCell Technologies | Cat#17898 | |
| Commercial assay or kit | EasySep Dead Cell Removal (Annexin V) Kit | StemCell Technologies | Cat#17899 | |
| Commercial assay or kit | Agilent Seahorse XF Cell Mito Stress Test Kit | Agilent Technologies | Cat#103015-100 | |
| Software | GraphPad Prism software | GraphPad (https://graphpad.com) | RRID:SCR_002798 | Version 9.3.1 |
| Software | Gen5 software | Biotek Instruments (https://www.agilent.com/) | | Version 3.05 |
| Software | Seahorse Wave Desktop Software 2.6 | Agilent Technologies (https://www.agilent.com/) | RRID:SCR_014526 | Version 2.6 |
| Software | ImageJ software | ImageJ (http://imagej.nih.gov/ij/) | RRID:SCR_003070 | |

## Cell culture and reagents

CAMA-1 (ATCC HTB-21) and MCF7 (ATCC HTB-22) human BC cell lines were maintained in Dulbecco's Modified Eagle Medium (DMEM) supplemented with 10% fetal bovine serum (FBS, Sigma-Aldrich #12306C) and antibiotic–antimycotic solution. T47D (ATCC HTB-133) human BC cell line was maintained in RPMI supplemented with 10% FBS and antibiotic–antimycotic solution. Cells were regularly

tested for mycoplasma contamination using commercially available Mycoplasma detection kit (Myco Alert kit, Lonza). Cell lines were authenticated using STR profiling (Laragen, Inc). Everolimus (RAD001, #S1120) and ONC201/TIC10 (#S7963) were obtained from Selleckchem and dissolved in DMSO.

Everolimus resistant cell lines were generated by long-term culture of parental cell lines in the continuous presence of 100 nM everolimus (MCF7 and T47D) or 50 nM everolimus (CAMA-1) with fresh media and drug replenished every 3 days, until resistance developed (6–12 months). Resistance to everolimus was confirmed by the difference in the drug dose–response in comparison with the parental cells, measured using CellTiter-Glo Luminescent Cell Viability Assay (Promega Corporation #G7573). Everolimus resistant cells were further maintained in complete culture medium supplemented with 50 nM everolimus (CAMA-1) or 100 nM everolimus (MCF7 and T47D). Sensitive and resistant cells were labeled with Venus (LeGO-V2 Addgene #27340) and mCherry (LeGO-C2 Addgene #27339) fluorescent proteins, respectively, as described previously (*Grolmusz et al., 2020*; *Weber et al., 2008*).

## Malignant fluid collection and primary cancer cell isolation

Malignant fluids were collected from five female BC patients by paracentesis (Patient # 1, 2, 4, and 5) or thoracentesis (Patient # 3) under informed consent and ethical compliance under Institutional Review Board (IRB) #07047 and #17334 at City of Hope. Selection criteria were determined from tumor type and treatment. Patient demographic data were not used for selection criteria. Upon collection of malignant fluid, cells were pelleted at 500 × *g* for 5 min, at room temperature. Red blood cells (RBCs) were then removed by lysis in Tris-ammonium chloride buffer (17 mM Tris, pH 7.4, 135 mM ammonium chloride) as previously described (*Nath et al., 2021*) or by magnetic depletion using EasySep RBC Depletion Reagent (StemCell Technologies) diluting cell pellet in 2% FBS in phosphate-buffered saline (PBS) and incubated with 50 µl Depletion Reagent/ml for 5 min at room temperature, followed by two rounds of incubation on EasyEight magnet (StemCell Technologies) for 5 min. Primary human BC cells were then purified by magnetic depletion using the EasySep Dead Cell Removal (Annexin V) Kit (StemCell Technologies #17899) and EasySep CD45 Depletion Kit II (StemCell Technologies #17898) to remove dead cells and immune cells, respectively, as previously described (*Nath et al., 2021*).

## Cell proliferation

For 2D experiments, cells were plated in 384-well plates and treated with increasing concentration of everolimus (0–100 nM) or ONC201/TIC10 (0–10 µM) or combination ONC201/TIC10 (0–10 µM) with everolimus (1 or 100 nM) for 4 days and viability was measured using CellTiter-Glo Luminescent Cell Viability Assay (Promega Corporation #G7573). Results were normalized to DMSO control treatment for each cell line. IC50 was determined using GraphPad Prism 9.3.1 software. Experiments were performed trice in quadruplicates and representative images are shown.

For 3D experiments, cells were plated in 96-well round-bottom ultra-low attachment spheroid microplate (Corning) at a density of 2000 cells per well (for CAMA-1 cells) or 5000 cells per well (for MCF7 and T47D cells). Spheroids were treated with drugs as indicated for up to 18 days with imaging and media change every 3–4 days. Imaging was performed using Cytation 5 imager (Biotek Instruments) gathering signal intensity from brightfield, YFP (for Venus fluorescence) and Texas Red (for mCherry fluorescence) channels. Raw data processing and image analysis were performed using Gen5 3.05 software as previously described (*Grolmusz et al., 2020*). Briefly, stitching of 2 × 2 montage images and Z-projection using focus stacking was performed on raw images followed by spheroid area analysis. Whole spheroid area and fluorescence intensity measurements of each cell line are integrated into a fitted growth equation, and cell counts for each cell line were produced from fluorescence intensities relative to spheroid size. Experiments were performed trice in triplicates and representative image is shown. Bliss Interaction Index was calculated as previously described (*Soldi et al., 2013*) with *I* > 1 showing synergy and *I* = 1 showing additivity. For 2D assay Bliss Interaction Index was calculated from luminescence values and for 3D studies from fluorescence values.

For 3D experiments with primary cells, cells were plated in 96-well round-bottom ultra-low attachment spheroid microplate (Corning) at a density of 20,000 cells per well in Renaissance Essential Tumor Medium (Cellaria) supplemented with RETM Supplement (Cellaria), 10% FBS, 25 ng/ml cholera toxin (Sigma-Aldrich) and antibiotic–antimycotic solution. After 2 days that spheroid structures were formed, spheroids were treated with drugs as indicated for 4 days. Brightfield imaging was performed

using Cytation 5 imager (Biotek Instruments) before treatment and after 4 days of treatment. Viability was measured using CellTiter-Glo 3D Cell Viability Assay (Promega Corporation #G9682). Results were normalized to DMSO control. Experiments were performed in triplicates and representative image is shown.

## Western blot

For immunoblot analysis, cells were washed with PBS and lysed on ice in ice-cold Radioimmunoprecipitation assay buffer (RIPA) buffer (Thermo Scientific) supplemented with protease and phosphatase inhibitor cocktail (Thermo Scientific). The protein concentration in the lysates was determined using BCA (Pierce). Equal amounts of total protein were separated by sodium dodecyl sulfate–polyacrylamide gel electrophoresis (SDS–PAGE) and 4–20% Tris-Glycine Gel (Bio-Rad) and were transferred to Polyvinylidene difluoride (PVDF) membranes using the iBlot Dry Blotting system (Invitrogen) according to the manufacturer's instructions. Membranes were immunoblotted overnight with antibodies against BiP (C50B12) (#3177, RRID:AB_2119845), pFoxO3a (Ser294) (#5538, RRID:AB_10696878), FoxO3a (D19A7) (#12829, RRID:AB_2636990), pERK1/2 Thr202/Tyr204 (197G2) (#4377, RRID:AB_331775), ERK1/2 (137F5) (#4695, RRID:AB_390779), pAKT Ser473 (D9E) (#4060, RRID:AB_2315049), AKT (C67E7) (#4691, RRID:AB_915783), pS6Ser240/244 Ribosomal Protein (D68F8) (#5364, RRID:AB_10694233), S6 Ribosomal Protein (54D2) (#2317, RRID:AB_2238583), β-actin (8H10D10) (#12262, RRID:AB_2566811), β-actin (13E5) (#5125, RRID:AB_1903890) from Cell Signaling Technology, CREB-2/ATF-4 (B-3) (sc-390063, RRID:AB_2810998), mtTFA (TFAM) (C-9) (sc-376672, RRID:AB_11150497) from Santa Cruz Biotechnology, TUFM (#PA5-27511, RRID:AB_2544987) and TRAIL (#MA1-41027, RRID:AB_1087999) from Thermo Fisher Scientific and CHOP (Cat#15204-1-AP, RRID:AB_2292610) from Proteintech. For the OXPHOS complexes detection, equal amounts of total protein were separated by SDS–PAGE and 16.5% Tris-Tricine Gel (Bio-Rad) and were transferred to PVDF membranes using the iBlot Dry Blotting system (Invitrogen) according to the manufacturer's instructions. Membranes were immunoblotted overnight with total OXPHOS Human WB Antibody Cocktail (ab110411, RRID:AB_2756818) from Abcam that contains 5 mAbs, against Complex I subunit NDUFB8, Complex II subunit 30 kDa, Complex III subunit Core 2, Complex IV subunit II, and ATP synthase subunit alpha. Quantitation of proteins was performed using ImageJ analysis. Experiments were performed thrice, and representative results are shown.

## Extracellular flux analysis

Mitochondrial respiration was determined using Agilent Seahorse XF Cell Mito Stress Test Kit (Agilent Technologies #103015-100) according to the manufacturer's instructions. Cells were plated in XF96 Cell Culture Microplate (Agilent Technologies Inc) coated with Cell-Tak adhesive (Corning) at a density of 15,000 cells per well. Six hours later cells were treated with drugs as indicated for 18 hr. Before running the assay, cell culture media were replaced with Seahorse XF assay media supplemented with 10 mM XF Glucose, 1 mM XF Pyruvate and 2 mM XF L-Glutamine and the plate was incubated in a 37°C incubator without $CO_2$ for 1 hr. The OCR was measured by XFe96 extracellular flux analyzer (Agilent Technologies) with sequential injection of 1.5 µM oligomycin A, 1 µM Carbonyl cyanide 4-(trifluoromethoxy)phenylhydrazone (FCCP), and 0.5 µM rotenone/antimycin A. For normalization, imaging was performed using Cytation 5 imager (Biotek Instruments) gathering signal intensity from brightfield, YFP (for Venus fluorescence) and Texas Red (for mCherry fluorescence) channels. Raw data processing and image analysis were performed using Gen5 3.05 software. Briefly, Gen5 3.05 software detected and generated cell counts and fluorescence intensity measurements respective for each cell line within each well. Data analysis was performed using Seahorse Wave Desktop Software 2.6. OCR was normalized to cell number. Experiments were performed trice in triplicates and representative results are shown.

## Gene expression data and pre-processing

Microarray expression data from a neoadjuvant clinical trial of ER+ BC patients on everolimus were retrieved from NCBI GEO (accession number GSE119262). The tumor samples were profiled using Illumina HumanRef-8 v2 Expression BeadChips and quantile normalized using BeadArray as described in *Sabine et al., 2010*. The normalized gene expression matrix was aggregated by averaging expression levels of probes mapping to the same gene and by averaging the expression levels of technical

replicates of the samples. RNA-seq data from MDA-MB-231 BC cell line treated with ONC201 were retrieved from NCBI GEO (accession number GSE212369). The samples were sequenced on the Illumina HiSeq 2500 platform, aligned to the h19 human genome reference using STAR, followed by gene level expression quantification using RSEM, as described in *Greer et al., 2018*.

## Pathway activity scores and analysis

Gene signatures for 4705 curated (C2) gene signatures were retrieved from the molecular signatures database (mSigDB version 7.0) (*Liberzon et al., 2011*). Single sample gene set enrichment scores were calculated from the normalized microarray (GSE119262) or RNA-seq (GSE212369) gene expression matrices using the R package GSVA (kcdf = 'Gaussian', method = 'ssgsea'). To identify pathway signatures that change in response to everolimus treatment, we analyzed the ssGSEA scores of the 21 pairs of samples from patients with gene expression data collected before and after treatment. We split the samples into responsive and non-responsive groups and identified the pathways with significant difference in mean ssGSEA scores between pre- and post-treatment groups using a paired *t*-test. We investigated the changes in pathway signatures in response to ONC201/TIC10 treatment using gene expression data from MDA-MB-231 cells treated with ONC201/TIC10 data collected at 0, 3-, 6-, 12-, and 24-hr time points. We obtained ssGSEA scores for the pathway signatures at each time point and fit a generalized linear model (gaussian family) for each pathway signature against time. After obtaining nominal p-values of the fit for all signatures, we calculated false discovery rates to identify signatures that changed significantly over time.

## Statistical analysis

Dose–response curves were generated using GraphPad Prism 9.3.1 software and statistical comparisons were performed using one-way analysis of variance. All data are presented as average values of samples, error bars correspond to standard deviation. For all other experiments, graphs were generated using GraphPad Prism 9.3.1 and statistical comparisons of the results were performed using Student's two-tailed *t*-test (*$p < 0.05$, **$p < 0.01$, ***$p < 0.001$, ****$p < 0.0001$).

## Acknowledgements

This study was supported by a US National Cancer Institute award number U01CA264620 and U54CA209978 awarded to AHB.

## Additional information

### Funding

| Funder | Grant reference number | Author |
|---|---|---|
| National Cancer Institute | U01CA264620 | Andrea H Bild |
| National Cancer Institute | U54CA209978 | Andrea H Bild |

The funders had no role in study design, data collection, and interpretation, or the decision to submit the work for publication.

### Author contributions

Elena Farmaki, Formal analysis, Investigation, Visualization, Methodology, Writing – original draft, Writing - review and editing; Aritro Nath, Data curation, Formal analysis, Visualization, Methodology, Writing – original draft; Rena Emond, Formal analysis, Investigation, Visualization, Methodology, Writing – original draft; Kimya L Karimi, Formal analysis, Investigation, Methodology; Vince K Grolmusz, Resources, Methodology; Patrick A Cosgrove, Resources, Methodology, Writing - review and editing; Andrea H Bild, Conceptualization, Resources, Supervision, Funding acquisition, Writing – original draft

### Author ORCIDs

Elena Farmaki ![ORCID] http://orcid.org/0000-0001-5038-5875

Patrick A Cosgrove (iD) http://orcid.org/0000-0001-7784-7785
Andrea H Bild (iD) https://orcid.org/0000-0003-4850-0453

## Ethics

Malignant fluids were collected from five female breast cancer patients by paracentesis (Patient # 1, 2, 4, and 5) or thoracentesis (Patient # 3) under informed consent and ethical compliance under Institutional Review Board (IRB) #07047 and #17334 at City of Hope.

## Decision letter and Author response

Decision letter https://doi.org/10.7554/eLife.85898.sa1
Author response https://doi.org/10.7554/eLife.85898.sa2

---

# Additional files

### Supplementary files
• MDAR checklist

### Data availability

The data that support the findings of this study are included within the manuscript. Source data files have been provided for Figure 4. All gene expression datasets analyzed in this study are publicly available. ER+ breast cancer patients' data are available from NCBI GEO (accession number GSE119262). RNA-seq data from MDA-MB-231 breast cancer cell lines are available from NCBI GEO (accession number GSE212369). Custom scripts used for the gene expression analyses are available on GitHub https://github.com/U54Bioinformatics/ONC201_Manuscript_2022 (copy archived at *Nath, 2022*).

The following previously published datasets were used:

| Author(s) | Year | Dataset title | Dataset URL | Database and Identifier |
|---|---|---|---|---|
| Sabine VS, Sims AH, Macaskill EJ, Renshaw L, Thomas JS, Dixon JM, Bartlett JM | 2010 | Gene expression profiling of response to mTOR inhibitor everolimus in pre-operatively treated post-menopausal women with oestrogen receptor-positive breast cancer | https://www.ncbi.nlm.nih.gov/geo/query/acc.cgi?acc=GSE119262 | NCBI Gene Expression Omnibus, GSE119262 |
| Greer YE, Porat-Shliom N, Nagashima K, Stuelten C, Crooks D, Koparde VN, Gilbert SF, Islam C, Ubaldini A, Ji Y | 2018 | ONC201 kills breast cancer cells in vitro by targeting mitochondria | https://www.ncbi.nlm.nih.gov/geo/query/acc.cgi?acc=GSE212369 | NCBI Gene Expression Omnibus, GSE212369 |

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
