## [Editor Report]

The study by Bild et al. reported a valuable finding on the combination use of ONC201/TIC10 towards ER+ breast cancer. The data of the manuscript is rather solid. The paper may produce translational impact and scientists working in the field of breast cancer may benefit most from the work.

---

## [Decision Letter]

**Decision letter after peer review:**

Thank you for submitting your article "ONC201/TIC10 enhances durability of mTOR inhibitor everolimus in metastatic ER+ breast cancer" for consideration by *eLife*. Your article has been reviewed by 2 peer reviewers, one of whom is a member of our Board of Reviewing Editors, and the evaluation has been overseen by Caigang Liu as the Senior Editor. The reviewers have opted to remain anonymous.

Essential revisions:

1) The authors should employ siRNA against ClpP to see if they can abrogate the biological effects of the drug combination.

2) Some cell viability assay experiments and Western experiments seem odd and therefore need to be re-conducted.

3) Possible toxicity issues should be discussed and elaborated in the main text.

*Reviewer #1 (Recommendations for the authors):*

Below are some specific points for the manuscript:

1. The cell viability curves for CAMA-1 seem a little odd. This experiment needs to be reconducted and amended.

2. Scale bars are required for Figure 2A.

3. Figure 3, Figure Legend, 'Viability', the first letter should always be capitalized and kept consistent in all figures.

4. Figure 4A-B, the bands for CHOP and ATF4 as well as a few others seem odd. This should be fixed.

5. Figure 4, ONCO201 (5uM/10uM), the authors need to quantify the bands to determine if the biological effects are dose dependent.

6. Supplemental Figure 2, scale bars are required.

7. Supplemental Figure 3, the bands for p-Foxo3a seem odd. This should be fixed.

8. References should be numbered and properly cited.

9. Again, some Western experiments should be reconducted and fixed as mentioned above. This is mandatory.

*Reviewer #2 (Recommendations for the authors):*

1. It would be nice to see whether you can abrogate the effects of the combination with siRNA against ClpP. That would strengthen the mechanistic point you make.

2. Please check in Figure 4 whether you show the β actin corresponding to all your blots.

3. A reflection on possible toxicity should be considered to be included in the Discussion section.

4. Please consider moving statements of interpretation of your data from the Results section to the Discussion section.

---

## [Author Response]

Essential revisions:Reviewer #1 (Recommendations for the authors):Below are some specific points for the manuscript:1. The cell viability curves for CAMA-1 seem a little odd. This experiment needs to be reconducted and amended.

We thank the reviewer for this valuable suggestion. Accordingly, we reconducted the dose response assays of CAMA-1, MCF7, and T47D everolimus sensitive and resistant cells with modified concentrations of everolimus that included lower doses (0-100nM) (Figure 1A). This figure has been updated in the manuscript.

2. Scale bars are required for Figure 2A.

We thank the reviewer for pointing out the low visibility of the scale bars in Figure 2A. We increased the resolution, thickness of the scale bars, and the font of letters and numbers and updated the images (Figure 2A).

3. Figure 3, Figure Legend, 'Viability', the first letter should always be capitalized and kept consistent in all figures.

We thank the reviewer for bringing this to our attention. This has now been edited (Figure 3A and B).

4. Figure 4A-B, the bands for CHOP and ATF4 as well as a few others seem odd. This should be fixed.

We thank the reviewer for this useful comment. We repeated the Western blot experiments for TUFM, TFAM, BiP, ATF4, CHOP, and β-actin for all 6 cell lines and treatments, using the iBlot Dry Blotting system (Invitrogen) for protein transfer on PVDF membrane to increase transfer efficiency for higher resolution figures. In addition, a different antibody was used for CHOP (CHOP Polyclonal Antibody Cat#15204-1-AP, RRID:AB_2292610, Proteintech) (Figure 4A and B). We believe these figures have increased resolution and have updated them in the manuscript.

5. Figure 4, ONCO201 (5uM/10uM), the authors need to quantify the bands to determine if the biological effects are dose dependent.

We thank the reviewer for this valuable suggestion. We performed quantitation of TUFM, TFAM, BiP, ATF4 and CHOP using ImageJ analysis. Protein expression levels were normalized to β-actin (Figure 4A).

6. Supplemental Figure 2, scale bars are required.

We thank the reviewer for pointing out the missing scale bars in Supplemental Figure 2. This has now been edited. (Figure 3—figure supplement 1).

7. Supplemental Figure 3, the bands for p-Foxo3a seem odd. This should be fixed.

We thank the reviewer for this useful comment. Similar to Comment 4, we reconducted the Western blot experiments for pAKT, AKT, pERK, ERK, pFoxO3a, FoxO3a, pS6, S6, TRAIL and β-actin for all 6 cell lines and treatments using the iBlot Dry Blotting system (Invitrogen) for protein transfer on PVDF membrane to increase transfer efficiency and resolution. This figure was updated in the manuscript. A different antibody was used for pFoxO3a (phospho-FoxO3a/Ser294 Antibody Cat#5538, RRID:AB_10696878, Cell Signaling) and TRAIL (TRAIL Monoclonal Antibody, 55B709.3 Cat# MA1-41027; RRID:AB_1087999, Thermo Fisher Scientific) (Figure 4—figure supplement 1).

8. References should be numbered and properly cited.

We have followed Journal guidelines for references.

9. Again, some Western experiments should be reconducted and fixed as mentioned above. This is mandatory.

We thank the reviewer for addressing this point. We reconducted the Western blot experiments for TUFM, TFAM, BiP, ATF4, CHOP, and β-actin for all 6 cell lines and treatments for Figure 4A and 4B, as well as the experiments for pAKT, AKT, pERK, ERK, pFoxO3a, FoxO3a, pS6, S6, TRAIL and β-actin for all 6 cell lines and treatments for Figure 4—figure supplement 1.

Reviewer #2 (Recommendations for the authors):1. It would be nice to see whether you can abrogate the effects of the combination with siRNA against ClpP. That would strengthen the mechanistic point you make.

We thank the reviewer for this suggestion. It has been previously reported in breast cancer that the mechanism of ONC201/TIC10 action involves binding and activation of ClpP through multiple silencing experiments (Graves et al., 2019; Greer et al., 2022) Our study is unique in the discovery that this therapy is effective in advanced ER+ breast cancer cells resistant to everolimus, a common standard of care drug used in this patient population. We show that ONC201/TIC10 has an increased therapeutic response in everolimus resistant cells compared to cancer cells sensitive to this therapy and detail the specific pathway components driving this response. Our study is the first to describe that ONC201/TIC10 could be used as an add-on treatment after mTOR therapy progression.

We also show that ONC201/TIC10 causes enhanced mitochondrial dysfunction, including suppression of mitochondrial proteins, oxidative phosphorylation inhibition and activation of stress response in everolimus resistant cells. In addition to the prior publication of the proposed knockdown studies (Greer et al., 2022; Jacques et al., 2020; Graves et al., 2019; Ishizawa et al. 2019), the siRNA knockdown methodology has limitations in our model system based on the duration of the siRNA-induced knockdown, as the 3-diminsional assays are measured over 3-4 weeks, and a transient effect from siRNA may not be sufficient over the duration of the experiment.

We have amended the manuscript to reference the previously established findings which describe how knockdown of ClpP abrogates the biological effects of ONC201/TIC10 (Discussion line 251, and references listed). We thank the reviewer for this suggestion, as it strengthens the link between drug and mechanism.

Lastly, for everolimus, the mechanism of action is well-known (Faivre et al., 2006; Law et al., 2006; Mita et al., 2003). Everolimus binds to the immunophilin FK Binding Protein-12 (FKBP-12) to generate a complex that inhibits the activation of the mammalian Target of Rapamycin (mTOR), a key regulatory kinase. As everolimus, an mTOR inhibitor, does not target ClpP (but targets mTOR signaling), the ClpP knockdown is not relevant for this drug’s mechanism. We have included this information in the manuscript for clarity.

Introduction section, line 34

“Everolimus, an analog of rapamycin, binds to the immunophilin FK Binding Protein-12 (FKBP-12) to generate a complex that inhibits the activation of mTOR, a key regulatory kinase (Faivre et al., 2006; Law et al., 2006; Mita et al., 2003).”

Discussion section, line 251

“Mechanistically, ONC201/TIC10 has been shown to directly activate ClpP, through various genetic and biochemical studies (Greer et al., 2022; Jacques et al., 2020; Graves et al., 2019; Ishizawa et al. 2019). In breast cancer, silencing of ClpP using siRNA or CRISPR/Cas9 System confirmed the ClpP-dependent effects of ONC201/TIC10 (Graves et al., 2019; Greer et al., 2022). Based on these established findings, we detail the specific pathway components driving the therapeutic response of ONC201/TIC10 in everolimus sensitive and resistant cells.”

References added:

Faivre, S., Kroemer, G., and Raymond, E. (2006). Current development of mTOR inhibitors as anticancer agents. Nat Rev Drug Discov 5, 671-688.

Graves, P.R., Aponte-Collazo, L.J., Fennell, E.M.J., Graves, A.C., Hale, A.E., Dicheva, N., Herring, L.E., Gilbert, T.S.K., East, M.P., McDonald, I.M., et al. (2019). Mitochondrial Protease ClpP is a Target for the Anticancer Compounds ONC201 and Related Analogues. ACS Chem Biol 14, 1020-1029.

Greer, Y.E., Hernandez, L., Fennell, E.M.J., Kundu, M., Voeller, D., Chari, R., Gilbert, S.F., Gilbert, T.S.K., Ratnayake, S., Tang, B., et al. (2022). Mitochondrial Matrix Protease ClpP Agonists Inhibit Cancer Stem Cell Function in Breast Cancer Cells by Disrupting Mitochondrial Homeostasis. Cancer Res Commun 2, 1144-1161.

Ishizawa, J., Zarabi, S.F., Davis, R.E., Halgas, O., Nii, T., Jitkova, Y., Zhao, R., St-Germain, J., Heese, L.E., Egan, G., et al. (2019). Mitochondrial ClpP-Mediated Proteolysis Induces Selective Cancer Cell Lethality. Cancer Cell 35, 721-737 e729.

Jacques, S., van der Sloot, A.M., C, C.H., Coulombe-Huntington, J., Tsao, S., Tollis, S., Bertomeu, T., Culp, E.J., Pallant, D., Cook, M.A., et al. (2020). Imipridone Anticancer Compounds Ectopically Activate the ClpP Protease and Represent a New Scaffold for Antibiotic Development. Genetics 214, 1103-1120.

Law, M., Forrester, E., Chytil, A., Corsino, P., Green, G., Davis, B., Rowe, T., and Law, B. (2006). Rapamycin disrupts cyclin/cyclindependent kinase/p21/proliferating cell nuclear antigen complexes and cyclin D1 reverses rapamycin action by stabilizing these complexes. Cancer Res 66, 1070-1080.

Mita, M.M., Mita, A., and Rowinsky, E.K. (2003). Mammalian target of rapamycin: a new molecular target for breast cancer. Clin Breast Cancer 4, 126-137.

2. Please check in Figure 4 whether you show the β actin corresponding to all your blots.

We thank the reviewer for the suggestion. We included images for all β-actin (Figure 4A and B).

3. A reflection on possible toxicity should be considered to be included in the Discussion section.

We thank the reviewer for this useful comment. We included a statement on possible toxicity in Discussion (Discussion section, line 301).

“Based on the initial preclinical and clinical testing of ONC201/TIC10 that demonstrated benefit with a benign clinical profile, ONC201/TIC10 is being further evaluated as single agent or in combination with other cancer therapies for various tumor types (Prabhu et al., 2020). Combination of ONC201/TIC10 with everolimus has been tested in an in vivo prostate model and was well tolerated, with no additional toxicity (Lev et al., 2018).”

4. Please consider moving statements of interpretation of your data from the Results section to the Discussion section.

We thank the reviewer for this remark. We updated the Results section and Discussion section accordingly.

Results section, line 171-172 moved to Discussion section, line 268-270

“This result suggests that in the sensitive cells, everolimus alleviates endoplasmic reticulum stress through the activation of different cell death mechanisms.”

Results section, line 187-188 moved to Discussion section, line 272-274

“Furthermore, Western blot analysis and gene expression data confirm that the observed activity of ONC201/TIC10 is via induction of mitochondrial stress rather than ERK/AKT inactivation.”

Results section, line 216-218 moved to Discussion section, line 234-236

“Our results show that non-responsive tumors continued to proliferate after everolimus treatment by utilizing oxidative phosphorylation pathway to fuel the growth of tumor cells and is consistent with the findings from the resistant cell lines, supporting that ONC201/TIC10 and modulation of mitochondrial function is effective in drug resistant cancer cells.”